# Characteristics of Electroencephalogram in the Prefrontal Cortex during Deep Brain Stimulation of Subthalamic Nucleus in Parkinson’s Disease under Propofol General Anesthesia

**DOI:** 10.3390/brainsci13010062

**Published:** 2022-12-29

**Authors:** Yuting Ling, Lige Liu, Simin Wang, Qianqian Guo, Qingyuan Xiao, Yi Liu, Bo Qu, Zhishuang Wen, Yongfu Li, Changming Zhang, Bin Wu, Zihuan Huang, Jianping Chu, Ling Chen, Jinlong Liu, Nan Jiang

**Affiliations:** 1Department of Anesthesiology, The First Affiliated Hospital of Sun Yat-sen University, Guangzhou 510080, China; 2Department of Neurosurgery, The First Affiliated Hospital of Sun Yat-sen University, Guangzhou 510080, China; 3Department of Neurology, The First Affiliated Hospital of Sun Yat-sen University, Guangzhou 510080, China; 4Department of Radiology, The First Affiliated Hospital of Sun Yat-sen University, Guangzhou 510080, China

**Keywords:** electroencephalogram, anesthesia, propofol, Parkinson’s disease, deep brain stimulation, subthalamic nucleus, processed EEG, microelectrode recording, intraoperative neuromonitoring

## Abstract

Background: Monitoring the depth of anesthesia by electroencephalogram (EEG) based on the prefrontal cortex is an important means to achieve accurate regulation of anesthesia for subthalamic nucleus (STN) deep brain stimulation (DBS) under general anesthesia in patients with Parkinson’s disease (PD). However, no previous study has conducted an in-depth investigation into this monitoring data. Here, we aimed to analyze the characteristics of prefrontal cortex EEG during DBS with propofol general anesthesia in patients with PD and determine the reference range of parameters derived from the depth of anesthesia monitoring. Additionally, we attempted to explore whether the use of benzodiazepines in the 3 days during hospitalization before surgery impacted the interpretation of the EEG parameters. Materials and Methods: We included the data of 43 patients with PD who received STN DBS treatment and SedLine monitoring during the entire course of general anesthesia with propofol in a single center. Eighteen patients (41.86%) took benzodiazepines during hospitalization. We divided the anesthesia process into three stages: awake state before anesthesia, propofol anesthesia state, and shallow anesthesia state during microelectrode recording (MER). We analyzed the power spectral density (PSD) and derived parameters of the patients’ prefrontal EEG, including the patient state index (PSI), spectral edge frequency (SEF) of the left and right sides, and the suppression ratio. The baseline characteristics, preoperative medication, preoperative frontal lobe image characteristics, preoperative motor and non-motor evaluation, intraoperative vital signs, internal environment and anesthetic information, and postoperative complications are listed. We also compared the groups according to whether they took benzodiazepines before surgery during hospitalization. Results: The average PSI of the awake state, propofol anesthesia state, and MER state were 89.86 ± 6.89, 48.68 ± 12.65, and 62.46 ± 13.08, respectively. The preoperative administration of benzodiazepines did not significantly affect the PSI or SEF, but did reduce the total time of suppression, maximum suppression ratio, and the PSD of beta and gamma during MER. Regarding the occurrence of postoperative delirium and mini-mental state examination (MMSE) scores, there was no significant difference between the two groups (chi-square test, *p* = 0.48; Mann–Whitney U test, *p* = 0.30). Conclusion: For the first time, we demonstrate the reference range of the derived parameters of the depth of anesthesia monitoring and the characteristics of the prefrontal EEG of patients with PD in the awake state, propofol anesthesia state, and shallow anesthesia during MER. Taking benzodiazepines in the 3 days during hospitalization before surgery reduces suppression and the PSD of beta and gamma during MER, but does not significantly affect the observation of anesthesiologists on the depth of anesthesia, nor affect the postoperative delirium and MMSE scores.

## 1. Introduction

Parkinson’s disease (PD) is a common degenerative neurological disease in the elderly, with clinical symptoms including motor and non-motor symptoms such as anxiety and sleep disorders [1,2]. Deep brain stimulation (DBS) of the subthalamic nucleus (STN) can effectively improve the motor symptoms and some non-motor symptoms of PD [3]. The key to the success of DBS is the precise location of the electrodes [4]. In STN DBS surgery, microelectrode recording (MER) plays an irreplaceable role in increasing the accuracy of the target location [5,6,7].

However, the high requirements for target localization in DBS surgery bring challenges to anesthesiologists, including, but not limited to, the necessity of patient cooperation, optimization of patient comfort, difficulty of airway management when patients are under the stereotactic head frame and away from the anesthesia field of vision, and the potential interference of anesthetics with MER [8,9,10]. Therefore, an optimized anesthesia regimen is essential so that a comfortable state of sedation or anesthesia can be achieved while minimizing the impact on MER. The feasibility and safety of STN DBS under propofol general anesthesia have been confirmed [11,12,13,14,15]. The success of general anesthesia intervention depends on the choice of anesthetic drugs, accurate administration of anesthetic drugs, and strict monitoring of the depth of anesthesia.

In common practice of general anesthesia STN DBS, shallow anesthesia is maintained during MER, first, by reducing the blood concentration of anesthetic drugs in advance, and then by adjusting the titration of anesthetic drugs and the depth of anesthesia along with an electroencephalogram (EEG) depth monitor and STN signal [13,14,15]. Therefore, monitoring the depth of anesthesia by scalp EEG based on the prefrontal cortex is a conventional and important means to achieve accurate control of the depth of anesthesia by STN DBS under general anesthesia. In terms of the advantages of this method, it can avoid inhibition of the STN signal during the operation due to anesthesia that is too deep, which affects the precise implantation of electrodes. Additionally, it can avoid complications such as body movement, cough, cardiovascular stress reaction, or cerebral hemorrhage caused by intracranial hypertension due to intraoperative awareness or shallow anesthesia [11]. A commercial EEG depth monitor obtains derived parameters through real-time analysis, which assists anesthesiologists to accurately judge patient awareness.

Nevertheless, the anesthesia depth monitor in asleep DBS remains in the clinical application stage, with no in-depth research yet conducted. It has been reported that the use of bispectral index (BIS), Narcotrend, or the newer SedLine for asleep STN DBS can realize anesthesia depth monitoring [11,14]. Under general anesthesia, the BIS is 40–60, Narcotrend is D-E, and the patient state index (PSI) from the SedLine monitor is 25–50 [16]. During MER, the available shallow anesthesia state is as follows: BIS is 60–80, Narcotrend is at the C-D level, and the maintenance level of PSI has not yet been reported in detail. Indeed, no previous study has analyzed the characteristics of prefrontal EEG and determined the reference range of parameters derived from anesthesia depth monitoring during STN DBS for patients with PD under propofol general anesthesia.

To study the characteristics and derived parameters of prefrontal EEG, we must consider the group specificity and individual differences of PD. Importantly, the EEG characteristics of patients of varying ages under anesthesia are different, while most patients with PD are elderly, which deviates from the age distribution characteristics of the population based on the calculation of the derived parameter algorithm [17]. Furthermore, PD is a degenerative disease of the central nervous system, in which the pharmacodynamic characteristics of anesthetics are abnormal. The dose of anesthetic that may be appropriate for general patients may be excessive for those with PD [18,19]. This is different from the EEG characteristics under propofol anesthesia based on the algorithm for calculating derived parameters. Finally, because patients with PD often have sleep disorders, such as insomnia, rapid eye movement sleep behavioral disorders, and others, benzodiazepine use is frequent in this population. Similar to most anesthetic and sedative drugs such as propofol, benzenediazepines act on γ- aminobutyric acid type A (GABA-A) receptors to influence the neurotransmitter transmission in the brain and induce sedation, hypnosis, and antianxiety effects [20,21]. However, whether this affects the EEG characteristics and derived parameters of patients with PD under anesthesia, thereby interfering with the anesthesiologist’s judgment of the depth of anesthesia, remains unclear. Therefore, it is important to study the application parameters and EEG characteristics of anesthesia depth monitoring in DBS under general anesthesia for the clinical work of DBS anesthesia.

In this study, we summarize the range of the derived parameters of the depth of anesthesia monitoring and the characteristics of the prefrontal EEG of patients with PD during STN DBS under propofol general anesthesia. We then explore the effect of benzodiazepines on the EEG characteristics. The aim of this study was to reveal the characteristics of prefrontal cortex neuron firing in patients with PD undergoing STN DBS under propofol anesthesia to guide the judgment of the depth of anesthesia, provide personalized programs for optimizing patients to receive asleep DBS, and ultimately achieve accurate regulation of anesthesia and improve patient prognosis.

## 2. Materials and Methods

### 2.1. Patient Population

This was a retrospective observational study conducted from September 2019 to April 2022, including patients with PD who received propofol full-course general anesthesia and bilateral STN DBS treatment and were monitored by SedLine in the First Affiliated Hospital of Sun Yat-sen University. The original EEG characteristics of the prefrontal cortex were analyzed. The study was approved by the relevant local ICE for clinical research and animal trials of the First Affiliated Hospital of Sun Yat-sen University (IIT2022-097). As we retrospectively analyzed routine clinical data required in anesthesia and surgery, the requirement for written informed consent from the patients was waived.

The inclusion criteria were as follows: (1) patients with PD who underwent bilateral STN DBS surgery under general anesthesia with propofol from September 2019 to April 2022; and (2) patients who had received SedLine monitoring (Masimo Corporation, Irvine, CA, USA). The exclusion criteria were as follows: (1) patients who had previously undergone brain-related surgery; (2) patients with incomplete preoperative evaluation data; (3) patients with other neurological disorders; and (4) patients with missing or incomplete EEG data. Our study included 63 people, among whom, two underwent brain surgery, one lacked non-motor evaluation data, and 17 had missing or incomplete EEG data (see Figure 1). Therefore, 43 people were included in this study. Eighteen patients had taken benzodiazepines in the 3 days’ hospitalization prior to surgery.

### 2.2. Surgical Procedures

During hospitalization, the patients were permitted to take benzodiazepines if needed. All antiparkinsonian medications were withdrawn overnight (at least 12 h) before surgery. Each patient was scanned using a 3.0 T magnetic resonance imaging (MRI) instrument to locate the STN target. On the day of the surgery, all patients were positioned in a stereotactic frame (Leksell G type) under local anesthesia to perform computed tomography (CT) scanning. After fusing the MRI and CT images, the VectorVision neuronavigational system was used to calculate the three-dimensional coordinates of the STN and plan out 10 electrode tracks.

As in previous studies [11,22], all DBS surgical operations were performed by the same surgeon. After scalp incision and perforation, a single-channel microelectrode with a 25 μm diameter was conducted by the automatic MicroGuide system (AlphaOmega Engineering, Nazareth, Israel). MER was used to determine the final position of the electrodes. Following completion of DBS lead (Model 3389 Medtronic, Inc.,Minneapolis, MN, USA, or PINS L302, Beijing PINS Medical Co., Beijing, China) insertion on both sides, the extension lead and pulse generator were immediately embedded. The patients were sent to the intensive care unit for monitoring until the head CT examination the next day did not indicate cerebral hemorrhage. The patient was discharged approximately 7 days after the operation, then returned to turn on the DBS program in a month’s time.

### 2.3. Anesthetic Procedures

On the day of the operation, after the establishment of peripheral venous fluid infusion channels, vital signs monitoring, invasive arterial blood pressure monitoring, and SedLine monitoring were performed. All patients received local scalp anesthesia with ropivacaine 0.5%. For induction, propofol was first administered by TCI (Alaris Medical Systems, Inc., San Diego, CA, USA) based on the Marsh model with a target plasma concentration of 4 μg/mL. The effect room concentration “x” μg/mL of propofol was recorded when the patient lost consciousness, and it was used as the minimum maintenance concentration during the follow-up MER. Remifentanil was then administered by TCI (Orchestra Base Primea, Fresenius Vial, Brezins, France) based on the Minto model with a target plasma concentration of 4 ng/mL and administration of cisatracurium (0.2 mg/kg). After endotracheal intubation, anesthesia was maintained with propofol TCI (2–3 μg/mL), remifentanil TCI (4 ng/mL), and cisatracurium (1.5 μg/kg/min). At least 10 min before MER, the target propofol and remifentanil concentrations were reduced from “x” to 1 μg/mL and 2 ng/mL, respectively. After MER, the pulse generator and extension lead were implanted under anesthesia with propofol, remifentanil, and sufentanil. During the whole operation, norepinephrine (0.01–0.1 µg/kg/min) was used to maintain the mean arterial pressure (MAP) at more than 60 mmHg.

### 2.4. Data Collection

Demographic data and baseline characteristics were obtained from the anesthesia and medical records, including age, sex, body mass index (BMI), education duration, age at onset of PD, duration of PD, Hoehn and Yahr stage, levodopa equivalent daily dose, comorbidity (hypertension or diabetes), and preoperative evaluation. The preoperative motor and non-motor evaluation was completed 1 week in advance by a designated physician trained in neurology. The motor evaluation included motor score (Movement Disorder Society–Unified Parkinson’s Disease Rating Scale (MDS-UPDRS)), motor type, and the result of the levodopa challenge test. The motor type was classified according to the MDS-UPDRS calculations proposed by Stebbins et al [23]. Freezing of gait was determined according to item 3.11 in the MDS-UPDRS. The non-motor evaluation included the mini-mental state examination (MMSE), Epworth sleepiness score (ESS), Pittsburgh sleep quality index (PSQI), Parkinson’s disease sleep scale (PDSS), Hamilton depression scale (HAMD), and Hamilton anxiety scale (HAMA). Apathy was determined according to item 1.5 in the MDS-UPDRS. Rapid eye movement sleep behavior disorder (RBD) was determined according to whether the patient exhibited shouting, punching, and kicking at night, as recorded in the preoperative electronic medical record. Preoperative prefrontal silent ischemia was based on the 3T-MRI report. Additionally, we recorded whether patients took benzodiazepines and antipsychotics before surgery. The Confusion Assessment Method (CAM) was used to evaluate delirium [24]. Delirium was assessed from the first postoperative day of discharge and confirmed by daily nursing, family, and accompanying personnel interviews.

### 2.5. EEG Data Acquisition and Processing

We divided the anesthesia process experienced by the patient into the following three stages: awake state before anesthesia, propofol anesthesia state, and shallow anesthesia state during MER. The MAP, heart rate, potential of hydrogen, carbon dioxide, blood glucose, and lactic acid of the three stages were recorded in detail. A SedLine monitor was used to obtain raw EEG data from the prefrontal electrodes. Specifically, after cleaning the forehead with an alcohol swab and slightly abrading the skin, the six-electrode array was applied at positions Fp1, Fp2, F7, and F8, with the ground at Fpz and the reference at AFz. The impedance was kept below 5 kΩ in each channel, and the A/D converter sampled at 178 Hz for all channels. Processed-EEG-derived parameters (PSI, electromyography (EMG), percentage of the suppression ratio, artifacts, spectral edge frequency (SEF) of the right cerebral hemisphere (SEF-R) and left cerebral hemisphere (SEF-L)) after continuous recording processing were obtained every 2 s. The monitor display parameters used were set at 10 μV/mm and 30 mm/s.

The raw EEG data were originally collected from the monitor using a USB port. We divided the data obtained by each patient into three files according to the corresponding time recorded in the three states. The data were respectively imported into MATLAB software (Matlab R2013b, Mathworks, Natick, MA, USA) and then preprocessed using the EEGLAB toolbox. The space coordinate positions of the four electrodes were manually entered. In the corresponding period, artifacts, such as blinking, EMG, and electrotome noise, were manually removed. The EEG time series of each artifact-free channel was then band-pass filtered between 0.1 Hz and 80 Hz, with a 50 Hz notch filter and the average reference. We used Welch’s method to estimate the average power spectral density (PSD) of the four channels of each patient (overlap rate 50%, time window 1 s). The obtained results were logarithmically transformed. We analyzed five power bands: slow-wave oscillation (SWO, 0.2–1.5 Hz), delta (1.6–3 Hz), alpha (8–12 Hz), beta (13–39 Hz), and gamma (40–80 Hz). The derived parameters exported by the monitor are presented in Excel. In situations where the artifact value was >50, the signal was regarded as noise or invalid information irrelevant to the brain and was not analyzed.

### 2.6. Statistical Analyses

All statistical analyses were performed with SPSS version 26.0 software (IBM Corp., Armonk, NY, USA). Continuous variables with a normal distribution are expressed as the mean ± standard deviation (SD), continuous variables with a non-normal distribution are expressed as medians (interquartile range), and categorical variables are expressed as numbers (percentage). The patients were divided into groups according to whether they took benzodiazepines within 3 days before surgery during hospitalization. The independent-samples t-test, Mann–Whitney U test, and chi-square test were used for comparisons between groups. Significance was recognized when *p* < 0.05. Linear regression analysis was conducted to exclude the influence of age and age of onset on the positive results.

## 3. Results

### 3.1. Baseline Characteristics

From September 2019 to April 2022, 43 patients underwent bilateral STN DBS surgery under general anesthesia with propofol-remifentanil and were monitored by SedLine in the First Affiliated Hospital of Sun Yat-sen University. The baseline characteristics, intraoperative vital signs, and blood gas analysis results are listed in Table 1 and Table 2. The mean age of the 43 analyzed patients with PD was 61.42 ± 8.87 years and 24 (55.81%) were male. The mean PD duration was 9.58 ± 4.50 years. Those who took benzodiazepines (*n* = 18) had poor preoperative sleep assessment results compared to those who did not (*n* = 25) in another group (PSQI score, *p* < 0.01; PDSS score, *p* < 0.01). The age at the time of surgery and age at onset of PD were statistically different between the two groups (both *p* = 0.01). To exclude the influence of vital signs and biochemistry on EEG results, we also listed the MAP, heart rate, oral temperature, potential hydrogen, arterial pressure of carbon dioxide, glucose, and blood lactic acid in the awake state before anesthesia, propofol anesthesia state, and shallow anesthesia state during MER. The results were normal and comparable between the two groups.

### 3.2. EEG Parameters

We showed the detailed range of EEG-derived parameters, including the PSI, EMG, total time of suppression, maximum suppression ratio, SEF-R, and SEF-L, in three states (see Table 3). For suppression during MER, there were significant differences between no benzodiazepine and benzodiazepine users in the total time of suppression and maximum suppression ratio (*p* = 0.01, *p* = 0.02, respectively; see Figure 2 and Appendix A for details). For PSI, SEF-R, and SEF-L, there was no significant difference in the three states between the two groups (see Figure 3).

### 3.3. PSD Results

Figure 4 shows the PSD data of one patient in three states. Figure 5 shows that the mean PSD was significantly different between the two groups in the beta and the gamma bands during MER (no benzodiazepine users vs. benzodiazepine users, mean ± SD; beta: −13.20 ± 2.65 vs. −10.92 ± 3.78, *p* = 0.03; gamma: −18.28 ± 2.38 vs. −16.24 ± 3.37, *p* = 0.03; also see Appendix A). There was no significant difference in the average PSD at different frequencies during the awake state and anesthesia states between no benzodiazepine and benzodiazepine users.

### 3.4. Linear Regression Analysis

The results of linear regression analysis are shown in Table 4. The results showed that the total time of suppression, maximum suppression ratio, and average PSD of beta and gamma during MER were only significantly correlated with the use of benzodiazepine, but were unrelated to age or age at onset of PD.

### 3.5. Clinical Outcomes

In the 7-day follow-up results, 1 incision was infected, 3 had transient hiccups, 3 had urinary incontinence, and 17 had delirium (39.53%). The MMSE scores decreased by 22.64 ± 5.08, but there was no statistical difference between the two groups.

## 4. Discussion

In this study, we retrospectively reviewed the data of 43 patients with PD undergoing bilateral STN DBS in a single center, under general anesthesia with propofol and monitored by SedLine. Our results showed three main advantages.

First, as in research on EEG, our data are true and objective given that they are based on real patients originating from clinical practice, instead of animal models or healthy volunteers. In addition to basic demographic characteristics, we considered clinical characteristics and other factors that may affect EEG for detailed screening as extended baseline, such as hemodynamic characteristics and blood gas analysis results. Moreover, this was a cohort study including elderly patients with neurodegenerative diseases who required brain surgery. We also explored the influence of benzodiazepines, which may affect the central nervous system, on the EEG characteristics and derived parameters. These factors are routinely excluded by other studies on anesthesia depth monitoring and EEG. Second, this is the first study to investigate the device parameter range of the anesthesia depth monitor and the PSD characteristics of prefrontal EEG during asleep DBS. Third, we determined that taking benzodiazepines in the 3 days prior to surgery did not affect the interpretation of the monitoring of the depth of anesthesia, but that the PSD of beta and gamma reduced under the light anesthesia state during MER, as would the occurrence of burst inhibition. The results also showed that this did not reduce the occurrence of postoperative delirium. Our results provide important information for surgeons and anesthesiologists, as well as a reference for the clinical work of anesthesiologists and surgeons.

For the awake state, we found that the average PSI of patients with PD was 89.86 ± 6.89. The PSI in a prospective observational study of healthy adult volunteers was 3 units lower (mean: 92.0) [25]. In contrast, the EMG and SEF were higher, while the suppression ratio was similar, which may be due to the elderly characteristics of patients with PD. Moreover, the age range of their study population was 23–63 years, which was far lower than that of the current study population. Furthermore, different age groups have different EEG characteristics [17]. Some studies have shown that with an increase in age, the relevant power changes from a lower frequency to a higher frequency, and the power of alpha is also gradually reduced [26]. This may just conform to the characteristics of high SEF and relatively low PSI. Additionally, in the awake state, the EEG we collected was under natural conditions, and there was no mandatory requirement for patients to keep their eyes open or closed, which may also have an impact. The PSI may cause an increase in EMG activity due to blinking and eyeball movement, thus interfering with the instrument’s algorithm analysis of the EEG waveform. More importantly, our research population may not be comparable to healthy adult volunteers. The ideal use object of the anesthesia depth monitor usually excludes the elderly, patients with central nervous system diseases, patients receiving central nervous system drugs, or those undergoing cerebral surgery. As this is a cohort of patients who are traditionally difficult to monitor with EEG, it is necessary to conduct independent research on their EEG.

The PSI range was determined in PD patients during general anesthesia with propofol. Under normal circumstances, the recommended depth of anesthesia is 25–50; during MER, we reduced the depth of anesthesia by reducing the concentration of anesthetic to avoid excessive interference to the electrical signal of the target. Our results provide potentially important reference suggestions for surgeons and anesthesiologists. It is appropriate to have a PSI range of 62.46 ± 13.08 during MER; the depth of anesthesia within this range is sufficient, and it is safe and feasible without excessive drug reduction or deepening of anesthesia, without affecting the electrical signal of the target, and without obvious complications such as intraoperative awareness.

Our study shows that taking benzodiazepines in the 3 days during hospitalization before surgery by patients with PD does not significantly impact the PSI results. Note that this is different from the sedative state taken in a short time before surgery. An earlier study described a mean baseline PSI of 80 in the conscious/sedative state 30 min before surgery with midazolam [27]. The influence of benzodiazepines on PSI was closely related to pharmacokinetics; in other words, patients who take benzodiazepines 30 min before surgery may lead to lower PSI, while taking drugs at night (at least 10 h) has almost no effect on the PSI results of the next day. Notably, the previous study excluded patients undergoing brain surgery, had a lower average patient age (41.1 years), only gave PSI results, and did not provide EMG and patient vital signs or other information as a reference.

Our results show that taking benzodiazepines at night before the operation in patients with PD can reduce the suppression ratio during MER under propofol anesthesia. Burst suppression on EEG indicates a serious decrease in neuronal activity and metabolic rate and activity; this is commonly seen in patients with hypothermia, hypoxia, drug overdose, coma, general anesthesia, and in patients with decreased basic levels of brain activity [28,29]. Recent evidence shows that the occurrence of burst suppression during general anesthesia is related to postoperative delirium and cognitive dysfunction [30,31,32]. However, the mechanism to explain this relationship remains unclear, and it is unknown whether postoperative psychosis can be reduced by reducing burst suppression. Furthermore, it is unclear whether the burst suppression can be reduced by reducing the amount of narcotic drugs. Our results show that reducing burst suppression does not reduce the incidence of postoperative delirium, and there was no significant difference in the postoperative cognitive function between the two groups. SedLine has been used in previous studies, which have reported that the commercial anesthesia depth index based on EEG cannot explain age-related brain changes [33]. This may be related to age-related changes in the brain structure and function [34,35]. Therefore, we used linear regression analysis to explain the impact of age on burst inhibition. We found that the reduction of burst inhibition was not related to age, but was related to the preoperative use of benzodiazepines. We speculated that the metabolic rate and activity of brain function under light anesthesia may become less active or dull due to sleep disorders or related neural pathways of benzodiazepines acting on STN through the GABA receptor. However, these considerations are speculative and require further study in future investigations. Although our results showed that allowing patients with PD to take benzodiazepines at night before surgery can reduce the burst suppression of the prefrontal cortex during MER, we cannot conclude that this was a protective behavior because it does not reduce the incidence of postoperative delirium. Of course, it cannot be ruled out that the low sample size may have led to no significant difference in the statistical results. Notably, burst suppression is considered an abnormal EEG mode that needs to be avoided. Therefore, from this perspective, whether the occurrence of postoperative delirium is reduced after surgery, taking benzodiazepines appropriately before surgery according to the patient’s sleep habits may have other potential benefits.

Our research has limitations. We did not strictly record the average consumption of propofol and remifentanil, nor did we provide the monitoring data of cerebral oxygen saturation, largely due to the retrospective nature of the study. For the former, our anesthesia scheme was strictly unified, so we can ensure that the target concentration of anesthetic drugs was consistent. As for the data of cerebral oxygen saturation, the MAP in different anesthesia stages may explain that the cerebral perfusion in each stage was sufficient without statistical difference. We will supplement these data as much as possible in the future. Finally, the sample size of this study was small, and further randomized, large sample size multicenter studies are needed to confirm our conclusions.

## 5. Conclusions

We have summarized the reference range of the derived parameters of the depth of anesthesia monitoring of the characteristics of the prefrontal EEG of patients with PD in the awake state, propofol anesthesia state, and shallow anesthesia during MER. It is appropriate to have a PSI range of 62.46 ± 13.08 during MER. The depth of anesthesia within this range is sufficient, and it is safe and feasible without excessive drug reduction or deepening of anesthesia, without affecting the electrical signal of the target, and without obvious complications such as intraoperative awareness. Taking benzodiazepines in the 3 days during hospitalization before surgery will reduce suppression and the PSD of beta and gamma during MER, but will not affect the observations of anesthesiologists on the depth of anesthesia, nor affect the postoperative delirium and MMSE scores. Further randomized, large sample size multicenter studies are needed to confirm our conclusions.

## Figures and Tables

**Figure 1 brainsci-13-00062-f001:**
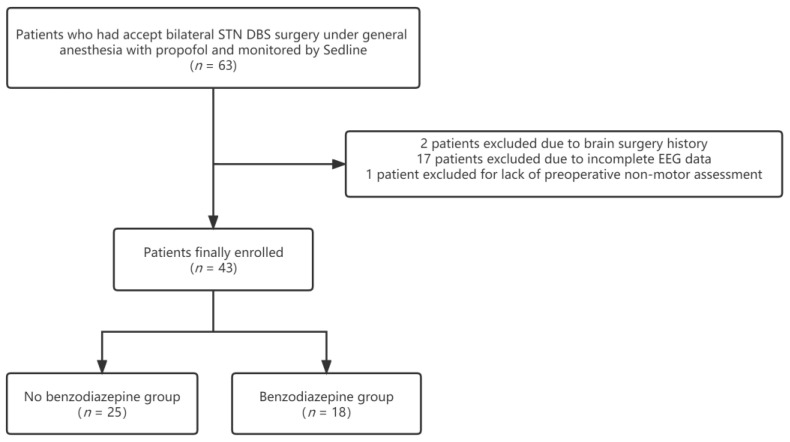
Flowchart detailing the selection of patients included in the retrospective analysis.

**Figure 2 brainsci-13-00062-f002:**
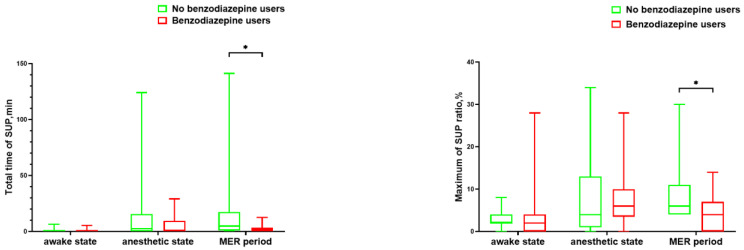
Box chart showing the suppression between the two groups during MER. For suppression during MER, there were significant differences between no benzodiazepine and benzodiazepine users regarding the total time of suppression and maximum suppression ratio. Significant changes are highlighted by * (*p* < 0.05). MER: microelectrode recording, SUP: suppression.

**Figure 3 brainsci-13-00062-f003:**
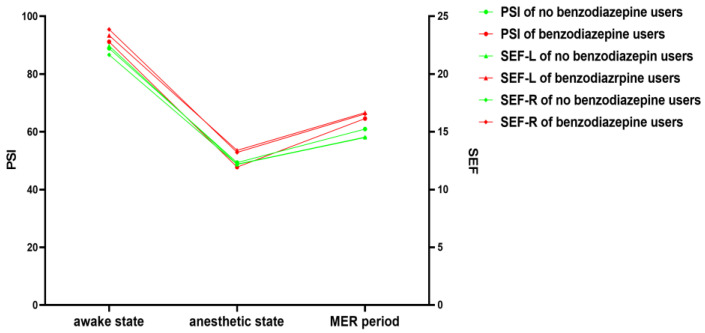
Trend chart reflecting the overall and derived parameters of the two groups. MER: microelectrode recording, PSI: patient state index, SEF-L: spectral edge frequency, left side, SEF-R: spectral edge frequency, right side.

**Figure 4 brainsci-13-00062-f004:**
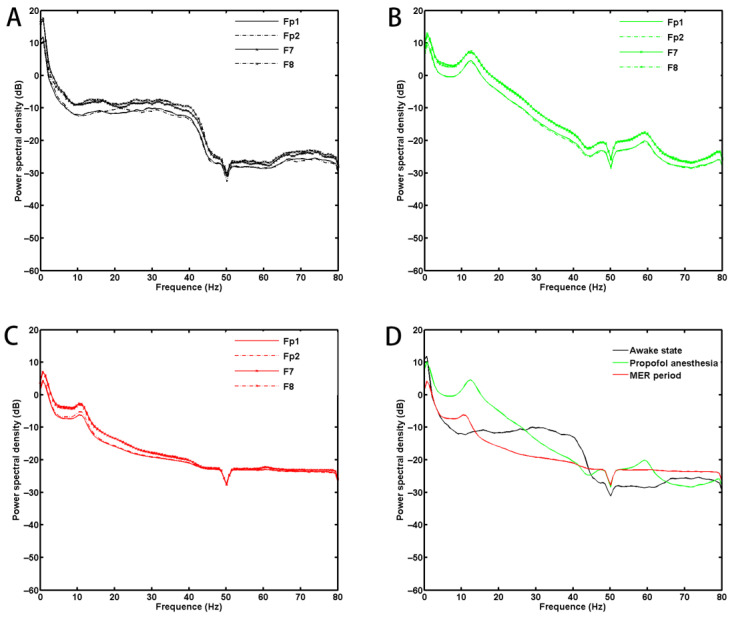
Power spectral density data of one patient in three states. Power spectral density of the four channels of one patient in awake states (**A**), propofol anesthesia state (**B**), and shallow anesthesia during MER (**C**). (**D**) shows power spectral density data of three states of a channel (Fp1).

**Figure 5 brainsci-13-00062-f005:**
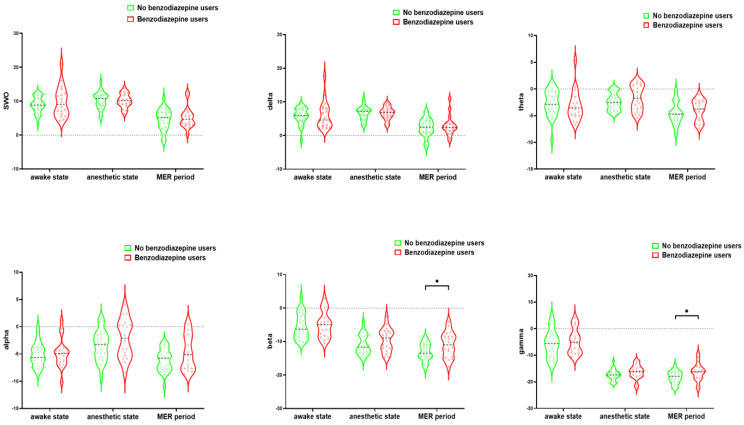
The violin diagram shows the average power spectral density of each frequency band in three states. The mean power spectral density was significantly different between the two groups in the beta and gamma bands during MER (no benzodiazepine users vs. benzodiazepine users, mean ± SD; beta: −13.20 ± 2.65 vs. −10.92 ± 3.78, *p* = 0.03; gamma: −18.28 ± 2.38 vs. −16.24 ± 3.37, *p* = 0.03). SWO (0.2–1.5 Hz), delta (1.6–3 Hz), alpha (8–12 Hz), beta (13–39 Hz), gamma (40–80 Hz). SWO: slow-wave oscillation, MER: microelectrode recording.

**Table 1 brainsci-13-00062-t001:** Baseline characteristics.

	Total		No Benzodiazepine Users		Benzodiazepine Users		*p*
	*n* = 43		*n* = 25		*n* = 18		
Age (years)	61.42	±8.87	58.72	±9.69	65.17	±6.01	0.01 ^a^
Male sex	24	(55.81)	16	(64.00)	8	(44.44)	0.23 ^b^
Body mass index (kg/m^2^)	23.49	±5.71	24.60	±6.44	21.95	±4.20	0.14 ^a^
Education duration (years)	10.77	±5.14	10.96	±5.07	10.50	±5.36	0.78 ^a^
Hypertension	7	(16.28)	3	(12.00)	4	(22.22)	0.43 ^b^
Diabetes	4	(9.30)	3	(12.00)	1	(5.56)	0.63 ^b^
Age at onset of PD (years)	51.88	±8.08	49.16	±7.91	55.67	±6.84	0.01 ^a^
Duration of PD (years)	9.58	±4.50	9.56	±5.33	9.61	±3.15	0.97 ^a^
Hoehn and Yahr stage, off	3.11	±0.69	2.98	±0.57	3.28	±0.83	0.20 ^a^
Hoehn and Yahr stage, on	2.20	±0.63	2.10	±0.58	2.33	±0.69	0.23 ^a^
Levodopa challenge test (%)	62.57	±13.79	65.74	±14.61	58.16	±11.52	0.08 ^a^
MDS-UPDRS IA	3.33	±2.67	2.80	±2.47	4.06	±2.84	0.13 ^a^
MDS-UPDRS IB + II	24.02	±11.52	21.08	±9.34	28.11	±13.21	0.05 ^a^
MDS-UPDRS III, off	45.33	±15.00	43.12	±12.60	48.39	±17.73	0.29 ^a^
MDS-UPDRS III, on	17.65	±10.30	15.00	±8.31	21.33	±11.84	0.05 ^a^
MDS-UPDRS IV	7.16	±4.04	6.88	±3.83	7.56	±4.40	0.60 ^a^
Motor type (tremor)	15	(34.88)	9	(36.67)	6	(33.33)	0.86 ^b^
MMSE score	26.47	±3.46	26.96	±3.20	25.78	±3.78	0.27 ^a^
ESS score	6.21	±5.54	6.20	±5.49	6.22	±5.77	0.99 ^a^
PSQI score	10.44	±4.80	8.36	±4.22	13.33	±4.07	<0.01 ^a^
PDSS score	99.73	±20.96	111.48	±15.84	83.42	±15.69	<0.01 ^a^
RBD	20	(46.51)	13	(52.00)	7	(38.89)	0.40 ^b^
Anxious	6	(13.95)	1	(4.00)	5	(27.78)	0.07 ^b^
Depressed	1	(2.33)	0	(0.00)	1	(5.56)	0.42 ^b^
Apathy	12	(27.91)	7	(28.00)	5	(27.78)	0.99 ^b^
Freezing of gait	24	(55.81)	11	(44.00)	13	(72.22)	0.06 ^b^
Prefrontal silent ischemia	23	(53.49)	11	(44.00)	12	(66.67)	0.14 ^b^
LEDD (mg)	897.91	±451.99	802.37	±259.05	1030.62	±614.75	0.15 ^a^
Antidepressants	12	(27.91)	5	(20.00)	7	(38.89)	0.17 ^b^
Anesthesia time (min)	429.19	±52.33	439.40	±51.70	415.00	±51.25	0.13 ^a^

Numbers indicate means ± standard deviations or number (percentage). The differences between the two groups were analyzed using a *t*-test (a) or chi-squared test (b). Significance was recognized when *p* < 0.05. PD: Parkinson’s disease, MDS-UPDRS: Movement Disorder Society–Unified Parkinson’s Disease Rating Scale, MMSE: mini-mental state examination, MOCA: Montreal Cognitive Assessment, ESS: Epworth sleepiness scale, PSQI: Pittsburgh sleep quality index, PDSS: Parkinson’s disease sleep scale, RBD: rapid eye movement sleep behavior disorder, LEDD: levodopa equivalent daily dose.

**Table 2 brainsci-13-00062-t002:** Comparison of vital signs and blood gas analysis results of the two groups of patients with Parkinson’s disease in three stages during deep brain stimulation in the subthalamic nucleus.

	N	Total		*n*	No Benzodiazepine Users		*n*	Benzodiazepine Users		*p*
Awake state before anesthesia										
Heart rate, bpm	43	81.27	±16.12	25	80.76	±16.21	18	81.97	±16.44	0.81 ^a^
MAP, mmHg	43	100.28	±14.16	25	102.61	±14.86	18	97.03	±12.82	0.21 ^a^
Oral temperature, °C	43	36.28	±0.37	25	36.30	±0.32	18	36.26	±0.44	0.69 ^a^
Potential of hydrogen	24	7.44	±0.03	16	7.44	±0.03	8	7.45	±0.03	0.96 ^a^
Carbon dioxide, mmHg	24	40.54	±4.93	16	40.50	±2.94	8	40.63	±7.82	0.95 ^b^
Glucose, mmol/L	24	5.80	±0.89	16	5.93	±1.05	8	5.54	±0.38	0.27 ^b^
Lactic acid, mmol/L	24	1.28	±0.65	16	1.36	±0.73	8	1.11	±0.46	0.39 ^a^
Propofol anesthesia state										
Heart rate, bpm	43	62.03	±9.39	25	62.03	±9.62	18	62.03	±9.32	0.99 ^a^
MAP, mmHg	43	76.91	±9.05	25	78.66	±9.54	18	74.47	±7.95	0.14 ^a^
Oral temperature, °C	43	36.17	±0.41	25	36.17	±0.39	18	36.17	±0.45	0.99 ^a^
Potential of hydrogen	23	7.43	±0.04	15	7.42	±0.04	8	7.45	±0.05	0.13 ^a^
Carbon dioxide, mmHg	23	40.13	±4.15	15	41.07	±4.20	8	38.38	±3.66	0.13 ^a^
Glucose, mmol/L	24	5.65	±0.80	16	5.69	±0.91	8	5.56	±0.55	0.81 ^b^
Lactic acid, mmol/L	24	1.25	±0.45	16	1.29	±0.49	8	1.15	±0.37	0.48 ^a^
MER period										
Heart rate, bpm	43	63.31	±11.19	25	63.00	±10.79	18	63.73	±12.02	0.84 ^a^
MAP, mmHg	43	79.07	±6.54	25	80.25	±6.55	18	77.43	±6.34	0.16 ^a^
Oral temperature, °C	43	36.04	±0.56	25	36.05	±0.59	18	36.02	±0.52	0.84 ^a^
Potential of hydrogen	24	7.44	±0.05	16	7.44	±0.05	8	7.46	±0.05	0.20 ^a^
Carbon dioxide, mmHg	24	38.54	±4.85	16	39.38	±4.99	8	36.88	±4.36	0.27 ^b^
Glucose, mmol/L	24	4.93	±0.65	16	4.96	±0.72	8	4.85	±0.51	0.70 ^a^
Lactic acid, mmol/L	24	1.95	±0.64	16	1.89	±0.51	8	2.08	±0.86	0.52 ^a^

Numbers indicate means ± standard deviations. Differences between the two groups were analyzed using a *t*-test (a) or Mann-Whitney U test (b). Significance was recognized when *p* < 0.05. MAP: mean arterial pressure, MER: microelectrode recording.

**Table 3 brainsci-13-00062-t003:** EEG parameters in patients with Parkinson’s disease in the three states during deep brain stimulation in the subthalamic nucleus (*n* = 43).

	Mean	SD	Median	Min	Max	P25	P75
Awake state before anesthesia							
PSI	89.86	6.89	92.51	69.80	97.73	87.94	94.37
SEF-L, Hz	22.81	4.14	24.13	9.95	28.24	19.88	26.29
SEF-R, Hz	22.58	4.13	23.17	12.21	28.79	19.88	26.15
Total time of SUP, min	0.85	1.50	0.35	0.00	6.50	0.05	0.67
Maximum SUP ratio, %	3.12	4.35	2.00	0.00	28.00	2.00	4.00
EMG, %	59.90	22.00	62.01	11.78	96.71	42.26	77.20
Propofol anesthesia state							
PSI	48.68	12.65	49.38	23.48	88.82	41.81	55.87
SEF-L, Hz	12.70	3.33	12.80	6.65	19.71	10.30	14.67
SEF-R, Hz	12.61	3.29	12.41	6.13	18.84	10.12	15.20
Total time of SUP, min	9.44	20.23	1.20	0.00	124.10	0.28	12.07
Maximum SUP ratio, %	7.49	7.38	6.00	0.00	34.00	2.00	10.00
MER period							
PSI	62.46	13.08	62.18	27.32	90.95	54.58	71.10
SEF-L, Hz	15.42	4.07	15.74	7.46	23.89	11.98	18.56
SEF-R, Hz	15.36	4.07	15.64	7.62	23.85	12.40	18.91
Total time of SUP, min	9.59	22.45	2.57	0.00	141.33	0.53	10.20
Maximum SUP ratio, %	7.12	6.25	6.00	0.00	30.00	4.00	10.00

EEG: electroencephalogram, SD: standard deviation, P25: 25th percentile, P75: 75th percentile, PSI: patient state index, SEF-L: spectral edge frequency, left side, SEF-R: spectral edge frequency, right side, SUP: suppression, EMG: electromyography, MER: microelectrode recording.

**Table 4 brainsci-13-00062-t004:** Results of linear regression analysis.

	**Total Time of SUP during MER**		**Maximum SUP Ratio during MER**	
	***B* (95% CI)**	** *p* **	***B* (95% CI)**	** *p* **
With/without benzodiazepine	−16.11 (−31.06, −1.16)	0.04	−5.19 (−9.21, −1.16)	0.01
Age, years	0.25 (−1.28, 1.77)	0.75	−0.23 (−0.64, 0.18)	0.27
Age at onset of PD, years	0.43 (−1.27, 2.14)	0.61	0.35 (−0.11, 0.81)	0.13
	**Average PSD of Beta during MER**		**Average PSD of Gamma during MER**	
	***B* (95% CI)**	** *p* **	***B* (95% CI)**	** *p* **
With/without benzodiazepine	2.33 (0.12, 4.54)	0.04	2.07 (0.09, 4.05)	0.04
Age, years	−0.05 (−0.28, 0.17)	0.64	−0.05 (−0.25, 0.15)	0.64
Age at onset of PD, years	0.04 (−0.21, 0.30)	0.73	0.04 (−0.18, 0.27)	0.71

Significance was recognized when *p* < 0.05. SUP: suppression, MER: microelectrode recording, PD: Parkinson’s disease, PSD: power spectral density.

## Data Availability

Research data are available upon request.

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
