# Peer review of "Characteristics of Electroencephalogram in the Prefrontal Cortex during Deep Brain Stimulation of Subthalamic Nucleus in Parkinson’s Disease under Propofol General Anesthesia"

_brainsci, 2022, doi:10.3390/brainsci13010062_

Round 1

Reviewer 1 Report

The paper is well written and of interest it contains 4 figures and 2 tables. 

The authors statistically analyzed EEG data from PD patients.

 conclusion: We have determined the reference range of the derived pa-41 rameters of the depth of anesthesia monitor and the characteristics of the prefrontal EEG of PD 42 patients in awake state, propofol anesthesia state and the shallow anesthesia during MER for the 43 first time. Taking benzodiazepines within 3 days before hospitalization will reduce suppression, the 44 power spectral density of beta and gamma during MER, but this will not affect the observation of 45 anesthesiologists on the depth of anesthesia, nor affect the postoperative delirium and MMSE 46 scores. 

Reviewer 2 Report

In Parkinson's disease, certain nerve cells (neurons) in the brain gradually break down or die. So, the topic selected by the authors is important and interesting.  Few points to be addressed:

(1) Discussion section is too long. If it will be point wise (like in bullets) then it will be more effective. 

(2) Conclusion section should be more details and clear.

(3) Abstract is ok.

(4) How full Introduction section is written without any references? Kindly give proper citation and include the references in the Reference Section.

(5) In Introduction Section, I suggest authors to include a paragraph to  mention the organization of the paper.

(6) Ref 1-15, 18-19 and 24-25 are not cited in the text.

Reviewer 3 Report

Authors extract the anesthesia monitor and the characteristics of the prefrontal EEG from the patients and showed meaningful data with the power spectral 464 density of beta and gamma during MER. English grammar and styles are very good. The detail analysis and research about the data would be good. Data analysis and patient categorization would be good. Authors showed good abstract section in detail. Authors also showed the research limitation. It is hard to find something wrong in entire manuscript. Therefore, the paper can be minor revision with following suggestive comments .

1. Supplementary Materials section need to be deleted if there is nothing.

2. Figures 2-4 label quality are so bad to be seen.

3. In Conclusion section, authors had better show future work.

4. Figure 5 label cannot be seen and label sizes are too small.

5. Author contribution section need to be checked with initial author names.

6. In Table 1, there is only male patient. Is there any female patient data ?

Round 2

Reviewer 2 Report

Thank you for addressing all the review comments.